# Price convergence in grain markets with seasonal differences

Michael Olabisi[1,2]*, Jiawen Liu[2], Toyin Ajibade[3], Mywish K. Maredia[2]

**1** Department of Community Sustainability (CSUS), Michigan State University, East Lansing, Michigan, United States of America, **2** Department of Agricultural, Food and Resource Economics (AFRE), Michigan State University, East Lansing, Michigan, United States of America, **3** Department of Agricultural Economics and Farm Management, University of Ilorin, Ilorin, Nigeria

These authors contributed equally to this work and share first authorship.

* olabism@msu.edu

## Abstract

Using weekly price data from 97 Nigerian markets, we examine how seasonal harvest timing shapes price dynamics for staple grains using a dyadic panel design. Our analysis reveals that markets operating in the same harvest phase experience faster price convergence, while asynchronous seasonal conditions slow adjustments— particularly for local rice and cowpea. In contrast, imported long-grain rice shows stable price behavior throughout the year. These results highlight the critical influence of seasonal cycles on market integration and offer fresh insights for food security strategies.

## Introduction

Price variations in food markets serve as a vital indicator of food security and market efficiency. In Nigeria, where evolving agricultural cycles and rapid urbanization influence the availability of staple grains, seasonal fluctuations in harvest timing play a central role in food price dynamics [1–4]. Changes in harvest periods create intervals of relative abundance and scarcity that affect both trading behavior and price convergence.

This paper focuses exclusively on seasonal price variations in Nigerian grain markets. We analyze weekly price data from 97 markets across northern and southern Nigeria, allowing us to compare price levels and gaps during harvest and non-harvest periods. By examining these temporal patterns, our study seeks to isolate the impact of crop cycles on price adjustments, thereby contributing to a more nuanced understanding of market integration under seasonal supply shocks [5,6].

The existing literature has documented the role of transportation costs, storage capabilities, and information flows in influencing price transmission [7–9]. While previous studies often addressed spatial price dispersion, our analysis concentrates solely on temporal dynamics. By emphasizing seasonal misalignment, we provide an

**Data availability statement:** The relevant data and code are available in a public GitHub repository at the following link: https://github.com/Joyce0077/Price-Convergence-in-Grain-Markets-with-Seasonal-Differences-replication-code.

**Funding:** MO - Grant Award Number 7200AA18LE00003. Funder Legume Systems Innovation Lab, for the United States Agency for International Development (USAID) https://www.canr.msu.edu/legumelab. MM - Same as MO above. The funders had no role in study design, data collection and analysis, decision to publish, or preparation of the manuscript.

insight into how variations in harvest timing affect price convergence, with important implications for food security policies and market stabilization efforts.

Our main contribution is to document how seasonal harvest timing shapes price convergence in grain markets. Using weekly price data from 97 markets in Nigeria, we show that price gaps are consistently smaller between markets that are in the same seasonal phase—particularly during harvest periods—while asynchronous conditions between harvest and non-harvest markets lead to slower price adjustment. These findings offer new insight into how seasonal dynamics influence market integration, extending the literature on food market efficiency in Africa [3,5,10]. They also complement the findings in recent papers on shock transmission within the global network of trade for food staples [e.g.,11,12].

This work complements research on market remoteness and volatility, which highlights the role of distance, infrastructure, and integration in shaping food prices [e.g.,13]. Seasonal misalignment, like spatial remoteness, contributes to price dispersion and weakens the transmission of supply shocks across regions. In addition, market access affects not only food prices but also the distribution of agricultural inputs and the adoption of improved technologies [6,14].

Our study relates to the literature on informXational and infrastructure frictions in food markets. While we do not directly measure information asymmetries, seasonal misalignment may function as a source of friction in itself, limiting arbitrage opportunities and reducing trade frequency during certain times of the year [7,8]. This is consistent with evidence that improved information flows reduce price gaps between markets [15].

Compared with papers that use monthly price data [e.g.,16], our use of high-frequency, weekly prices enables us to capture short-term fluctuations and responses to seasonal harvest cycles. The paper follows the methodological tradition of studies focused on price gaps and convergence [e.g.,5,17], while contributing new empirical evidence on how seasonal timing mediates integration in food markets. By emphasizing seasonality as a key dimension of price variation, the paper offers practical insights for designing regionally targeted interventions that enhance food security and market stability.

## 1 Theoretical framework

In this section, we sketch a simple framework that rationalizes the seasonal and spatial price patterns we estimate. Time is discrete $t = 0, 1, ...,$ and there are two locations $i \in \{A, B\}$ trading a storable staple. Let $P_{it}$ denote the (wholesale) price in market $i$ and $H_{it} \in \{0, 1\}$ indicate whether $i$ is in its post-harvest month ($H_{it} = 1$) or not ($H_{it} = 0$).

**Storage (intertemporal arbitrage).** With gross carrying cost $\Lambda \equiv 1 + r + c_s > 1$ (where $r \in [0, 1)$ is the interest rate and $c_s \in [0, 1)$ is the proportional storage cost, both expressed as fractions of value), the standard no-arbitrage condition implies

$$\mathbb{E}_t[P_{i,t+1}] \le \Lambda P_{it}, \qquad \text{with equality if inventories are positive.} \tag{1}$$

Thus, when stocks are carried over (the empirically relevant case immediately after harvest), expected prices rise over the marketing year at rate $\Lambda$, consistent with classic storage models [18–20].

**Spatial arbitrage (integration with trade costs).** Let iceberg trade cost between $i$ and $j$ be $\tau_{ij} \geq 0$. Competitive spatial arbitrage yields a price-band (law-of-one-price with frictions):

$$-P_{band} \equiv -\tau_{ij} \ \leq \ P_{it} - P_{jt} \ \leq \ \tau_{ij} \equiv P_{band}, \tag{2}$$

with trade flows such that $|P_{it} - P_{jt}| = \tau_{ij}$ when arbitrage is active and $|P_{it} - P_{jt}| < \tau_{ij}$ when markets are segmented [21, 22,22–24]. In our empirical analysis, bilateral trade costs $\tau_{ij}$ are proxied by distance and travel time, consistent with infrastructure-based approaches [25].

**Harvest shocks (within-market seasonality).** For intuition, consider two periods in a given market $i$: a harvest period $H$ and a non-harvest period $N$. In the harvest period, output is $Q_i^H$, and a competitive producer chooses how much to carry into the next period, denoted $S_i \in [0, Q_i^H]$. Let inverse demand in each period be locally linear, $P_i = A_i - B_i q_i$ with $B_i > 0$, where $q_i$ is quantity available for consumption. The harvest-period price is then

$$P_i^H = A_i - B_i(Q_i^H - S_i), \tag{3}$$

since $Q_i^H - S_i$ is consumed immediately.

In the non-harvest period there is no new production, so supply comes entirely from storage, implying

$$P_i^N = A_i - B_i S_i. \tag{4}$$

With gross carrying cost $\Lambda \equiv 1 + r + c_s > 1$ as in (1), the no-arbitrage condition for a competitive trader is $P_i^N \leq \Lambda P_i^H$, with equality when $S_i > 0$. In the empirically relevant case with positive inventories, this pins down

$$P_i^N = \Lambda P_i^H > P_i^H, \tag{5}$$

so the non-harvest price must exceed the harvest price to compensate for storage costs.

**Domestic varieties.** For domestic staples, let inverse demand be locally linear and write the reduced-form price around a reference quantity as

$$P_{it} \ = \ \alpha_i \ - \ \beta \, H_{it} \ + \ u_{it}, \qquad \beta > 0, \tag{6}$$

where $H_{it} = 1$ (a harvest month) shifts contemporaneous supply out and lowers price by $\beta$, and $u_{it}$ captures idiosyncratic shocks. Eq (6) delivers the familiar "low at harvest, then rising" seasonal pattern when combined with (1).

Subtracting (6) between locations gives

$$P_{it} - P_{jt} \ = \ (\alpha_i - \alpha_j) \ - \ \beta \, (H_{it} - H_{jt}) \ + \ (u_{it} - u_{jt}). \tag{7}$$

When harvest timing is *synchronous* ($H_{it} = H_{jt}$), the harvest component cancels out, and the cross-market gap is driven only by fixed differences ($\alpha_i - \alpha_j$) and shocks, remaining within the band (2) for a given $\tau_{ij}$. When timing is *asynchronous* ($H_{it} \neq H_{jt}$), the gap shifts outwards by roughly $\beta$, making $|P_{it} - P_{jt}|$ larger and more likely to hit the band, i.e., increasing the scope for segmentation when $\tau_{ij}$ is large and compressing only when arbitrage is sufficiently cheap. Immediately after a *joint* harvest, both markets carry inventories, so by (1) their expected price drifts are parallel; common drift mechanically dampens dispersion dynamics, reinforcing tighter convergence in the synchronous case.

**Imported varieties.** For imported (or globally priced) staples, local prices satisfy

$$P_{it} = P_t^* + \tau_i + \varepsilon_{it}, \tag{8}$$

where $P_t^*$ is the world price and $\tau_i$ a local access cost. Subtracting (8) between $i$ and $j$ yields

$$P_{it} - P_{jt} = (\tau_i - \tau_j) + (\varepsilon_{it} - \varepsilon_{jt}), \tag{9}$$

so that common movements in the world price cancel out and the cross-market gap is driven mainly by time-invariant access cost differences and idiosyncratic shocks. Because imported varieties are priced off the world market rather than local supply conditions, their cross-market gaps should exhibit muted seasonality and much weaker responses to local agricultural cycles—consistent with our long-grain rice results.

**Testable predictions.** Eqs (2)–(9) deliver three predictions we take to the data: (i) cross-market price gaps are smaller when harvest timing is synchronous than when asynchronous, holding $\tau_{ij}$ fixed; (ii) the effect of (a)synchrony on gaps attenuates with better integration (lower $\tau_{ij}$) and for imported varieties. Our reduced-form estimates with distance and travel-time controls are consistent with these predictions.

## 2 Data and descriptive statistics

Our study uses four primary data sources: price data, market network data, market pairs distance data, and crop growing season data.

*Price survey data*: Our main data source is a comprehensive set of weekly prices collected from 97 diverse markets in Nigeria, including 82 rural and 15 urban markets. The collection of markets spans both the northern and southern regions of the country, selected to capture the key link-markets that connect farms to consumers in Nigeria - by using a snowball survey approach from the known largest grain markets. We tried to conduct a census of all the grain sellers in each market. The aggregate of responses from each market serves as an estimate of its size. The market sizes range from 20 to 418, with an average of 85 responses.

We started the price data collection effort in February 2022 and continued through April 2023. The price variations captured cover the entire annual cycle, providing us with rich and detailed data. Related articles describe the approach to data collection and cleaning [26], as well as the setup for the survey [27]. Specifically, post-collection, we filtered out extreme values of reported prices. The remaining data were then averaged to get a representative market price per product for each market-week.

Our survey covered a variety of agricultural commodities, specifically cowpeas, local and long-grain rice, millet, sorghum, groundnuts, soybeans, maize, wheat, cassava, yam, and potatoes. In the subsequent analysis, we primarily focus on cowpeas, local rice, and long-grain rice, as they represent the predominant crops in our survey and in the Nigerian diet. The full list of products in the survey includes the most widely consumed food grain items in Nigeria.

Table 1 presents summary statistics for the prices of the top three main food items in our survey data: Cowpea, Local Rice, and Long-Grain Rice. Several patterns in the table fit expectations. Local rice is less expensive than long-grain rice (which is usually imported), while the seasonal price variation is nontrivial.

For cowpea, prices are higher during the harvest period, with an average of 467.17 NGN/kg compared to 435.76 NGN/kg during non-harvest periods. The price gap between markets when both are in the harvest season is 67.78 NGN/kg, and when both are in the non-harvest season it is 68.42 NGN/kg. In contrast, market pairs that mix harvest and non-harvest periods exhibit a larger average gap of 82.27 NGN/kg. This suggests that synchronizing harvest periods across markets leads to tighter price convergence for cowpea, whereas asynchronous seasonal conditions contribute to greater price dispersion.

**Table 1**. Summary statistics for product prices and price gaps.

| Variable | Cowpea | Local Rice | Long-Grain Rice |
|---|---|---|---|
| No. of Market Observations | 97 | 97 | 90 |
| Average Price | 456.38 | 489.90 | 583.14 |
| Average Prices Harvest | 467.17 | 558.32 | 627.80 |
| Average Prices Non-Harvest | 435.76 | 439.72 | 545.18 |
| Average Price Gap (Harvest to Harvest) | 67.78 | 55.79 | 86.56 |
| Average Price Gap (Harvest to Non-Harvest) | 82.27 | 108.22 | 76.29 |
| Average Price Gap (Non-Harvest to Non-Harvest) | 68.42 | 88.67 | 64.26 |

Prices are averages in NGN per kg. The most common measure for selling grains in wholesale markets is the 100kg bag, so most of the prices reported above can be understood as wholesale prices when multiplied by 100.

Price gaps too are shown in NGN/kg, $|P_{ikt} - P_{jkt}|$.

Local rice displays a similar seasonal pattern in terms of average prices, with a harvest period price of 558.31 NGN/kg and a non-harvest price of 439.72 NGN/kg. However, the price gaps differ markedly. When both markets are in the harvest phase, the average gap is relatively small at 55.79 NGN/kg. Conversely, when one market is in the harvest season and the other is not, the gap widens substantially to 108.22 NGN/kg, and even during non-harvest periods the gap remains high at 88.67 NGN/kg. These figures imply that seasonal alignment plays a critical role in the efficiency of price transmission for local rice, with asynchronous conditions leading to significantly greater price disparities.

Long-grain rice, which is predominantly imported, shows a distinct pattern. Although prices during the harvest period are higher (627.80 NGN/kg) than during non-harvest periods (545.18 NGN/kg), the differences in price gaps across seasonal categories are less consistent. The harvest-to-harvest gap is 86.56 NGN/kg, which is higher than the harvest-to-non-harvest gap of 76.29 NGN/kg, and the gap during non-harvest periods is the smallest at 64.26 NGN/kg. This pattern suggests that long-grain rice prices are less responsive to domestic seasonal cycles, likely reflecting the influence of international market forces on its pricing.

Some of the relatively larger gaps for harvest-to-non-harvest market pairs may also reflect geographic separation, as many such pairs connect northern and southern markets that are far apart. To account for this, our main regressions control for both physical distance and travel time between markets. Table 2 summarizes these transport-cost measures, which serve as key controls in isolating the effect of seasonal asynchrony from that of market separation.

*Market-to-market relationships*: A novel contribution of this research project to the literature is our network data on market-to-market relationships. The data on relationships between these 97 markets come from a trader survey conducted in 2022 where we asked traders in each market to name the markets where they source or send their products. The responses were coded as follows: [1] a market pair was coded as having a *direct* relationship if a trader in one market named a partner in the other market, [2] a market pair was coded as having an *indirect* relationship if there were no reported direct connections between the two markets, but there is at least one market with direct connections to both markets, [3] a market pair was coded as having an *direct and indirect* connections if the two types of connections link them, and [4] the last category covers markets with *no connections*—This category includes market pairs that do not have any direct or indirect connections through exactly one intermediary market. For our sampled 97 markets, we identified 676 connected market pairs—the 285 directly linked market pairs from the original data, and 491 pairs with indirect connections.

The network of market-to-market relationships, represented in Fig 1, uses different color codes for markets in northern and southern Nigeria.

The network reveals both dense intra-regional connections within the North and South, and substantial inter-regional linkages connecting the two regions. Many northern markets act as large hubs, with multiple connections extending toward the South, while southern markets, though smaller on average, also maintain key northbound trade routes. For

**Table 2.** Summary statistics for distance, travel time between market pairs.

| Connection | Average Price Gap | Distance (*km*) | Travel Time (*h*) | Obs |
|---|---|---|---|---|
| No connection | 0.18 | 570.85 | 9.27 | 286410 |
| Indirect Connection only | 0.15 | 524.72 | 8.45 | 15687 |
| Direct Connection only | 0.12 | 250.74 | 3.84 | 917 |
| Both Indirect and Direct Connection | 0.14 | 280.01 | 4.74 | 9770 |

Note: 1. The price gap is determined by taking the logarithms of absolute value of difference of the same commodity's prices index in two distinct markets. This value is then expressed as a ratio of the two prices, providing a relative measure of the price difference between these markets. 2. The price index is calculated by dividing the average monthly price by the yearly median price, enabling a standardized measure of price variation over time. 3. Distance and Travel time are computed via google distance api.

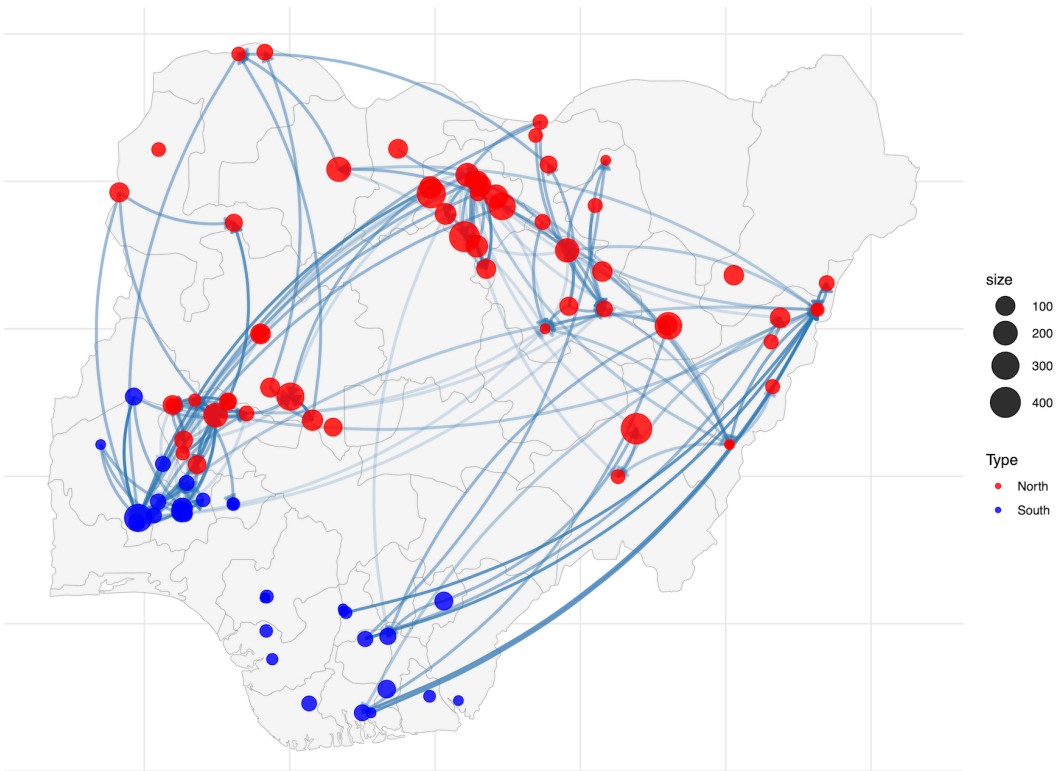

**Fig 1.** **Market linkages.** Note: The map plots the 97 surveyed markets, classified into northern (red) and southern (blue) regions (61 northern, 36 southern). Nodes represent markets, and edges indicate reported trading links between them. Edge transparency reflects the number of reported trade relationships, while node size corresponds to market size. Distinct node colors differentiate regions. Self-loops are excluded for clarity.

clarity, we must explain that our snowball approach started in Northern Nigeria, in the largest grain markets. A notable omission from the south-west part of the country is Lagos–the nation's most populous city, due to budget limitations. We expect that the omission will not change the substance of our main findings, given the city's proximity and strong connections to Bodija market (which we include, and which prior work puts forward as a key source of food grain price movements in the Ibadan-Lagos area) [28]. It is also important to clarify that the network graph only shows connections among the key markets for which we track prices. Where the only direct reported connections for a market are to seasonal or local markets, the market appears as an unconnected node in the graph.

*Distances and travel times*: Table 2 presents a summary of the data for each market pair, including the distances and travel times between the 97 markets featured in our price survey. We calculated the travel time and distance for all 9,312 possible pairs (97x96) of these markets, using the Google Distance API. This approach highlights the API's potential as an effective tool for assessing transportation costs or road quality, supplementing traditional methods of regional integration measurement, as noted in other studies [e.g.,29,30]. Previous papers largely relied on the absolute physical distance for such assessments [e.g.,13,31]. The richer data allows us to control for factors linked to road conditions and traffic, leading to a more accurate evaluation of transportation costs or road quality. This approach offers a stark contrast to other studies which simply used the proportion of asphalt-surfaced roads as an indicator of road quality [8]. The travel time data matters because it is a salient but less-studied proxy for the cost of moving goods between markets—an important determinant of market prices. We matched the distance and travel times to the respective product-week summaries, to get the unit of observations for our regression analysis, grouped by the reported connections between the market pairs,

Table 2 shows that price gaps are higher on average, and distances/travel times are higher too between markets with no reported connections. The average differences in prices indexes between market pairs in any given week is about 20% ($exp^{0.19}$). Market pairs with direct and indirect connections have smaller price gaps, compared with those without. As expected, distances and travel times are lowest for market pairs with direct connections, compared to those with indirect or both direct and indirect connections (about 5 hours on average). The number of observations reported in the last column of Table 2 is informative - as it shows fewer than 1,000 observations with direct connections in our data (representing 0.3% market-pairs). Most market-pairs have no reported connections at all. Just about 8% of the observations are for market pairs with either only indirect connections, or both indirect and direct connections (representing 4.9% and 3.1% market-pairs respectively).

*Growing seasons*: We incorporated the crop growing seasons data from the crop calendar dataset of the Center for Sustainability and the Global Environment at the University of Wisconsin-Madison. The dataset provides typical growing seasons for a broad range of crops across different geographical locations. It includes most of the crops present in our price survey data. For the few that were not included, such as cowpeas, we relied on the expert knowledge and traditional agricultural practices prevalent in Nigeria. It is important to note that the growing seasons can vary significantly based on geographical locations, as is evident in the case of Nigeria. For instance, the harvesting season for groundnut begins around August 1 in southern Nigeria, while it starts two months later, around October 1, in the northern region. Thus, our analysis accounts for such regional differences in crop seasons to accurately correlate them with the price data.

In Fig 2, we show price patterns using a three-panel graph, providing insights into the dynamics of detrended price indices for different commodities. The detrended price index is calculated by dividing the average monthly price by the yearly median price, providing a standardized measure of price variation over time. On the y-axis, we depict this detrended price index, while the x-axis represents the number of months since the end of the first (rainy-season) harvest month. Nigeria's rice harvest occurs in two distinct seasons—rainy and dry—but the x-axis is aligned only to the rainy-season harvest to ensure a consistent time reference across markets, as the dry-season harvest occurs in fewer locations and roughly six months later.

In the first facet, focusing on cowpea, we observe a decreasing trend in the price index until approximately six months after the end of the harvest. Subsequently, there is a notable increase in prices, indicating a shift in market dynamics. Moving to local rice, we find that prices experience a drop immediately after the end of the harvest, followed by a gradual increase, suggesting changing supply and demand dynamics for this commodity. In the context of long-grain rice—a variety predominantly imported into Nigeria—price fluctuations are not as distinct compared to locally grown rice. Prices peak during the harvest season and remain relatively stable outside this period, plausibly driven by holiday-season demand.

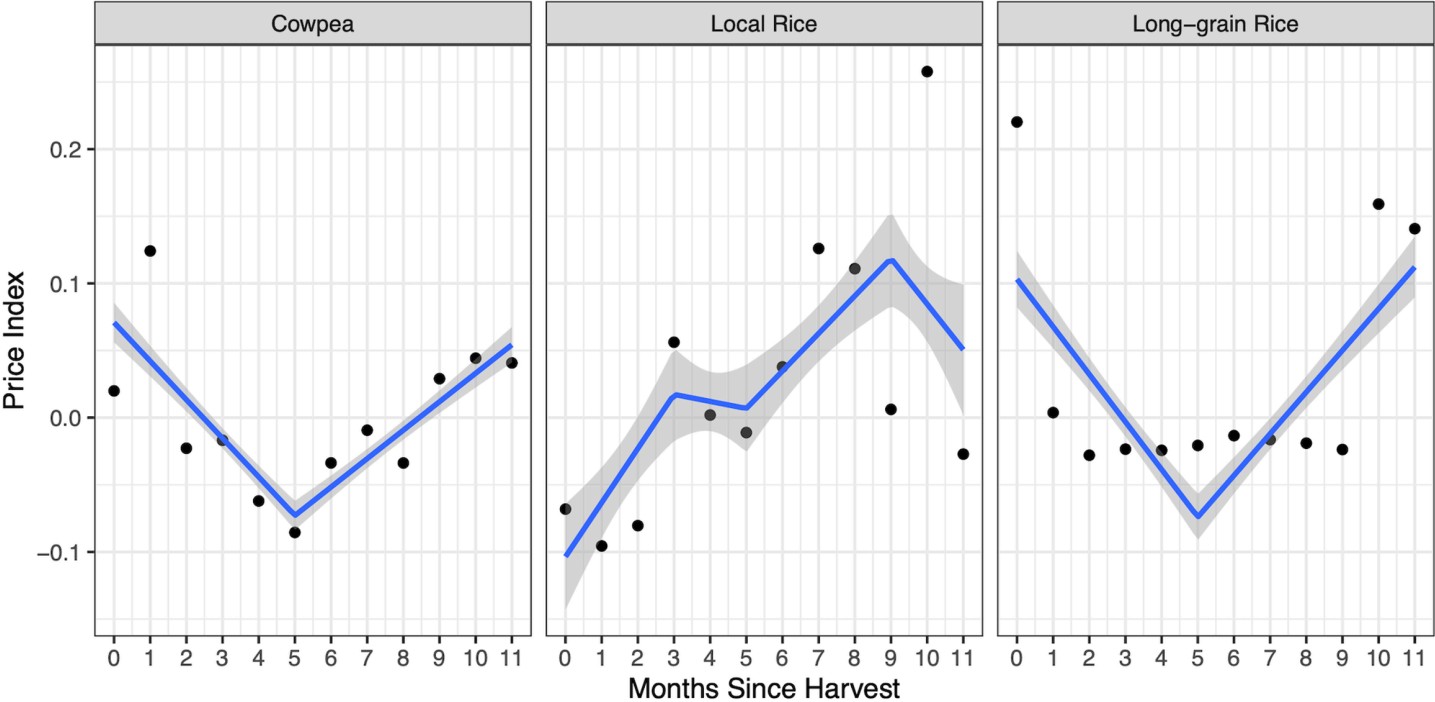

**Fig 2**. **Seasonal price variation by item—our market survey data (Feb 2022–Apr 2023).** Note: The dots on the graph show monthly average detrended price indices since harvest. Trend lines through the dots illustrate the trajectory of price changes over time, and the grey shaded region indicates the 95% confidence interval of these trends, demonstrating the potential variability of price movements.

## 3 Methods and results

### 3.1 Methods

For this paper, we follow the dyadic panel regression estimates in [10].

For each market-pair and product combination, we define the price-gap variable as

$$\text{PriceGap}_{ijt} = \log(P_{it}) - \log(P_{jt}),$$

which captures the logarithmic difference in prices between market $i$ and market $j$ at time $t$.

Our analysis focuses on the short-run dynamics of price convergence, emphasizing the role of seasonal harvest timing. In each week, market pairs are classified into one of three seasonal categories: Harvest to Harvest (both markets are in the harvest season), Harvest to Non-Harvest (one market is in the harvest season while the other is not), and Non-Harvest to Non-Harvest (neither market is in the harvest season). These seasonal statuses are incorporated into our dyadic panel regression model through interactions with the lagged price-gap variable.

Our baseline specification is:

$$\Delta\text{PriceGap}_{ijt} = \alpha + \sum_{s\in\{HH,HN,NN\}} \beta_s \left( D_{ijt}^s \cdot \text{PriceGap}_{ijt-1} \right) + \delta\, X_{ijt} + \mu_{ij} + \gamma_t + u_{ijt}. \tag{10}$$

where $\Delta\text{PriceGap}_{ijt}$ denotes the first difference of the price gap, $\text{PriceGap}_{ijt-1}$ is the lagged price gap, and $D^{HH}_{ijt}$, $D^{HN}_{ijt}$, and $D^{NN}_{ijt}$ are dummy variables indicating the seasonal category of the market pair. The term $\delta X_{ijt}$ incorporates an additional control variable, and $\mu_{ij}$ represents market pair fixed effects to account for unobserved, time-invariant heterogeneity between market pairs. Time fixed effects $\gamma_t$ capture common shocks across all markets, and $u_{ijt}$ is the error term.

This specification enables us to directly assess how seasonal misalignments in harvest timing affect the speed of price adjustments. We follow the dyadic panel regression framework outlined in [10] to estimate these relationships.

*Data ethics:* In collecting data for the estimation methods described above, we took care to ensure the following: The study was reviewed and approved by the MSU institutional review board (in Jan 2020) before the study began to contact survey respondents in 2022. Participants provided informed consent, with their verbal consent documented in the survey forms. No minors were included in the study.

## 3.2 Results

### 3.2.1 Seasonal differences in market integration.
To better illustrate the spatial variation in harvest timing across Nigeria, Appendix S3 presents the start and end dates of the harvest season for the key crops across different regions. The timing of harvest varies significantly between the northern and southern regions due to climatic differences, which influence planting and harvesting schedules.

This regional variation in harvest timing implies that market pairs involving inter-regional trade are more likely to fall into the "Harvest to Non-Harvest" category at some point in the year. This is particularly relevant for local rice, where harvesting takes place from August to November in the south but from November to January in the north. Similarly, cowpea harvests in the south precede those in the north, meaning that market pairs spanning both regions will frequently involve one market in harvest and the other not.

Given this variation, it follows that a substantial share of weeks will involve trade between markets in different harvest periods. The observed slower price adjustment in "Harvest to Non-Harvest" pairs may therefore be driven by these seasonal misalignments, where supply shocks in one region do not immediately translate into price adjustments in another. This underscores the importance of harvest timing in price dynamics and market integration.

The regression results in Table 3 confirm that price adjustment speeds vary significantly across seasonal trade relationships, reinforcing the role of harvest timing in market integration.

For Local Rice, Brown Cowpea, and White Cowpea, market pairs in the "Harvest to Harvest" category exhibit the fastest price convergence, suggesting that when both markets are in the harvest phase, prices adjust more rapidly due to high trading activity, increased supply, and stronger arbitrage opportunities. In contrast, the "Harvest to Non-Harvest" category shows the slowest price adjustments, particularly for Local Rice. This result aligns with the expectation that when one market is in harvest and the other is not, price convergence slows due to asymmetries in supply availability and demand responses. The harvest market experiences downward price pressure due to abundant supply, while the non-harvest market maintains relatively stable prices, causing a temporary misalignment that slows adjustment.

Interestingly, for Brown and White Cowpea, price adjustment is slowest in the "Non-Harvest to Non-Harvest" category, suggesting that trading volumes may decline outside the harvest period, leading to less frequent transactions and delayed price convergence. This pattern suggests that seasonal inactivity, rather than supply-demand asymmetry, could be the dominant factor limiting price adjustments when both markets are out of the harvest season.

Unlike locally produced crops, Long-Grain Rice exhibits no significant differences in price adjustment across seasonal categories. This is expected because Long-Grain Rice is primarily imported, meaning that its price dynamics are more influenced by other factors than domestic harvest cycles. Since supply is not seasonally constrained within Nigeria, price adjustments occur consistently throughout the year, reinforcing the idea that imported goods follow a different pricing mechanism than locally harvested crops.

 

**Table 3**. Adjustment speed estimates for connected markets by season.

| | Local Rice | Long-grain Rice | Brown Cowpea | White Cowpea |
|---|---|---|---|---|
| Dependent Var.: | $\Delta y_{ijt}$ | $\Delta y_{ijt}$ | $\Delta y_{ijt}$ | $\Delta y_{ijt}$ |
| Harvest to Harvest x $y_{ijt-1}$ | -1.057*** | -0.750*** | -0.890*** | -0.996*** |
| | (0.157) | (0.152) | (0.092) | (0.101) |
| Harvest to Non-Harvest x $y_{ijt-1}$ | -0.382*** | -0.597** | -0.785*** | -0.820*** |
| | (0.076) | (0.180) | (0.084) | (0.093) |
| Non-Harvest to Non-Harvest x $y_{ijt-1}$ | -0.603*** | -0.686*** | -0.658*** | -0.498*** |
| | (0.080) | (0.080) | (0.066) | (0.060) |
| Test: | | | | |
| Harvest to Harvest = Harvest to Non-Harvest | P=0.000 | P=0.562 | P=0.177 | P=0.014 |
| Harvest to Harvest = Non-Harvest to Non-Harvest | P=0.002 | P=0.681 | P=0.002 | P=0.000 |
| Harvest to Non-Harvest = Non-Harvest to Non-Harvest | P=0.000 | P=0.595 | P=0.068 | P=0.000 |
| Fixed-Effects: | | | | |
| Market pair | Yes | Yes | Yes | Yes |
| week | Yes | Yes | Yes | Yes |
| Controls | Yes | Yes | Yes | Yes |
| Observations | 598 | 593 | 875 | 1,002 |
| R2 | 0.708 | 0.419 | 0.597 | 0.537 |
| Within R2 | 0.582 | 0.125 | 0.367 | 0.393 |

Note: (1) . p<0.1; *p<0.05; **p<0.01; ***p<0.001. (2) This analysis includes only market pairs with direct or indirect connections, based on the premise that segregated markets lack mechanisms for price adjustment, rendering them unsuitable for market integration analysis. (3) Control variables include interaction term of $y_{ijt-1}$ with travel time, travel distance, and if markets are in the region. (4) Standard errors clustered by market-pairs.

## 4 Limitations and future research

This study has several limitations. First, our network data on market connections are not crop-specific, which may introduce noise in defining tradable market pairs. The bias is likely limited given that cowpea and local rice are available in all surveyed markets, and long-grain rice in most, and because these connections are used only to restrict the sample rather than as regressors. Second, while weekly price data enable fine-grained seasonal analysis, the 14-month coverage (February 2022–April 2023) may not capture longer-term structural patterns, macroeconomic shocks, or policy shifts. Finally, unobserved local factors—such as temporary supply disruptions, transport issues, or demand changes—may still influence price convergence, even among connected markets. Future research with crop-specific network data, longer time series, and richer contextual variables could deepen and extend the insights presented here.

## 5 Conclusion

Our analysis underscores the pivotal role of seasonal harvest timing in shaping grain price dynamics across Nigeria. We find that price convergence between regional markets is strongest when both markets are in harvest, as simultaneous surges in local supply drive prices down in unison. By contrast, convergence is markedly slowest when harvest timing is misaligned – when one region is reaping a bumper crop while another awaits its harvest, price disparities persist longer before leveling. This seasonal pattern of integration highlights that what might appear as spatial price friction is, in fact, often a temporal misalignment in production cycles.

These findings carry important implications for region-specific policy and food security planning. Nigeria's diverse agro-ecological zones harvest staple grains at different times, so a one-size-fits-all market policy is suboptimal. Instead, policies should be calibrated to local cropping calendars. For instance, in regions entering the post-harvest glut, programs might encourage grain storage or facilitate trade to deficit areas, whereas regions awaiting harvest might benefit from strategic grain reserves or timely imports to stabilize prices. Aligning policy interventions with harvest timing can help smooth out extreme seasonal price swings, protecting consumers in deficit areas from spikes and safeguarding farmers in surplus areas from gluts. In short, strengthening inter-regional market linkages during misaligned harvest periods – by

improving information flows and incentivizing inter-seasonal trade – can enhance food availability and affordability when and where it is most needed.

Finally, by focusing exclusively on seasonal production cycles, this study refines the literature on market integration and seasonal price variation in sub-Saharan Africa. Prior research has long noted that seasonal price volatility is high in African food markets [32–34] and that market integration can vary across the harvest cycle. Our contribution adds nuance to these insights by pinpointing harvest synchrony as a key factor: when harvests are synchronized between regions, markets integrate more swiftly, and when they are staggered, integration temporarily weakens. Recognizing this temporal dimension of market integration offers a more granular understanding of Africa's seasonal price patterns. It emphasizes that improving food security is not just about boosting production or infrastructure in general, but also about timing market interventions to the agricultural calendar. By accounting for when harvests occur in different regions, policymakers and development practitioners can better foster price convergence, mitigate seasonal scarcity, and enhance the overall stability of grain markets in Nigeria and beyond.

## Supporting information

**S1 Appendix. Robustness checks using FEWS NET price data.**
(PDF)

**S2 Appendix. Transportation cost and spatial price variation.**
(PDF)

**S3 Appendix. Harvest timing by crop and region in Nigeria.**
(PDF)

**S4 Appendix. Maddala–Wu unit-root test results for crop prices.**
(PDF)

**S5 Appendix. Inclusivity in global research.**
(DOCX)

## Author contributions

**Conceptualization:** Michael Olabisi.

**Data curation:** Toyin Ajibade.

**Formal analysis:** Michael Olabisi, Jiawen Liu.

**Funding acquisition:** Michael Olabisi, Mywish K. Maredia.

**Methodology:** Jiawen Liu.

**Project administration:** Michael Olabisi.

**Supervision:** Michael Olabisi, Toyin Ajibade, Mywish K. Maredia.

**Writing – original draft:** Michael Olabisi, Jiawen Liu.

**Writing – review & editing:** Michael Olabisi, Jiawen Liu, Toyin Ajibade, Mywish K. Maredia.

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
