## [Decision Letter · Decision Letter 0]

17 Jul 2025

PONE-D-25-30518

Price Convergence in Grain Markets with Seasonal Differences

PLOS ONE

Dear Dr. Olabisi,

Thank you for submitting your manuscript to PLOS ONE. After careful consideration, we feel that it has merit but does not fully meet PLOS ONE’s publication criteria as it currently stands. Therefore, we invite you to submit a revised version of the manuscript that addresses the points raised during the review process.

We look forward to receiving your revised manuscript.

Kind regards,

Eugenio Sebastian Antonio Bobenrieth,

Academic Editor

PLOS ONE

2. In the ethics statement in the Methods, you have specified that verbal consent was obtained. Please provide additional details regarding how this consent was documented and witnessed, and state whether this was approved by the IRB.

 [MO - Grant Award Number 7200AA18LE00003

Funder Legume Systems Innovation Lab, for the United States Agency for International Development (USAID)

https://www.canr.msu.edu/legumelab

MM - Same as MO above].

4. In the online submission form, you indicated that [The data underlying the results presented in the study are available from the authors to academic researchers on request.].

5. Please update your submission to use the PLOS LaTeX template. The template and more information on our requirements for LaTeX submissions can be found at http://journals.plos.org/plosone/s/latex"

Additional Editor Comments:

Reviewer 1:

This is a clearly-written and straightforward article using agricultural price data collected from 97 markets in Nigeria to show that locally-produced crops (rice and cowpea) experience faster price convergence across markets when both markets are in the harvest season. The analysis is based on reduced-form correlations. It would benefit from a better theoretical discussion of price behavior under storage costs (which explain low harvest prices that then increase throughout the year) and trade costs (e.g. under what conditions markets are segmented vs. trading) as this would help us understand the results and their implications. However, I think that is probably beyond the scope of this short article.

Here are some minor comments that I think should be addressed:

1) Please provide more information on the markets. How were they selected? Can you show them on a map? How many are in the South and how many in the North? (This seems to be the key division in terms of your harvest seasons in table 5.)

2) You should reference the table 2 results in the main paper. One of the reasons why the Harvest to Non-Harvest price gaps appear larger in table 1 is that markets with different harvest seasons (North vs. South) are further apart.

3) P. 13 / table 3: if I am reading the table correctly, the “Harvest to Non-Harvest” category is NOT the slowest for White Cowpea as you state in the text. It is only the slowest for Rice.

4) Appendix 1.a.: you should show the comparison that you did with the FEWS NET data. Also, please explain what is shown in figures 2 and 3 – is this your data or the FEWS NET data?

5) Typos to fix: The first sentence of section II has a double “and”; in table 1 the last row should be non-harvest to non-harvest.

6) Your statement that the data are available from the authors to academic researchers upon request does not satisfy this journal’s data policy as I understand it. You should make the data publicly available to all in an online repository.

----

Reviewer 2:

1. Originality and significance of contribution to existing knowledge

This paper investigates the impact of seasonal harvest timing on price convergence for staple grains across 97 Nigerian markets. The authors leverage weekly price data to argue that market integration is not merely a function of spatial factors but is critically mediated by the temporal synchrony of production cycles. Nevertheless, the empirical narrative largely confirms what the arbitrage condition already predicts (i.e., larger contemporaneous supplies compress gaps, asynchronous supplies widen them).

2. Appropriateness of the abstract as a description of the paper

The abstract clearly states the objective, dataset (weekly prices from 97 markets), and headline findings (faster convergence in synchronous harvests, slower in asynchronous ones). However, it provides no hint of the methodological framework.

3. Adequacy of the literature review

The authors successfully position their work within the existing literature on market integration, price transmission, and food security in Africa. The manuscript correctly references foundational and contemporary studies, effectively differentiating its temporal focus from prior work centered on spatial frictions. Yet the review leans heavily on sources published before 2020.

4. Organization and readability

The paper is logically structured, following a standard format that is easy to follow. The writing is generally clear. In Table 1, the variable "Average Price Gap (Harvest to Harvest)" is duplicated. Also, the authors reveals that the x-axis from Figure 1 is based only on the rainy season harvest, but does not explain why.

5. Soundness of methodology, analysis, and interpretation

The core methodological approach is suited to the research question. However, there are some concerns:

- the manuscript first states that “all data are fully available without restriction” and later that data “are available from the authors on request”; this is a contradiction;

- the authors acknowledge that their network data on market connections is not crop-specific, which introduces potential noise into the analysis;

- weekly prices offer granularity, yet the sample spans only 14 months (Feb 2022–Apr 2023); one harvest cycle could be insufficient to assess structural seasonal effects, macro shocks, or policy interventions;

- standard unit-root and cointegration tests should precede the regression;

6. Evidence supports conclusions.

The main conclusions are supported by the regression results. However, the paper lists no explicit limitation section.

Reviewers' comments:

Reviewer's Responses to Questions

**Comments to the Author**

1. Is the manuscript technically sound, and do the data support the conclusions?

Reviewer #1: Yes

Reviewer #2: Yes

2. Has the statistical analysis been performed appropriately and rigorously?

Reviewer #1: Yes

Reviewer #2: Yes

3. Have the authors made all data underlying the findings in their manuscript fully available?

Reviewer #1: No

Reviewer #2: No

4. Is the manuscript presented in an intelligible fashion and written in standard English?

Reviewer #1: Yes

Reviewer #2: Yes

5. Review Comments to the Author

Reviewer #1: This is a clearly-written and straightforward article using agricultural price data collected from 97 markets in Nigeria to show that locally-produced crops (rice and cowpea) experience faster price convergence across markets when both markets are in the harvest season. The analysis is based on reduced-form correlations. It would benefit from a better theoretical discussion of price behavior under storage costs (which explain low harvest prices that then increase throughout the year) and trade costs (e.g. under what conditions markets are segmented vs. trading) as this would help us understand the results and their implications. However, I think that is probably beyond the scope of this short article.

Here are some minor comments that I think should be addressed:

1) Please provide more information on the markets. How were they selected? Can you show them on a map? How many are in the South and how many in the North? (This seems to be the key division in terms of your harvest seasons in table 5.)

2) You should reference the table 2 results in the main paper. One of the reasons why the Harvest to Non-Harvest price gaps appear larger in table 1 is that markets with different harvest seasons (North vs. South) are further apart.

3) P. 13 / table 3: if I am reading the table correctly, the “Harvest to Non-Harvest” category is NOT the slowest for White Cowpea as you state in the text. It is only the slowest for Rice.

4) Appendix 1.a.: you should show the comparison that you did with the FEWS NET data. Also, please explain what is shown in figures 2 and 3 – is this your data or the FEWS NET data?

5) Typos to fix: The first sentence of section II has a double “and”; in table 1 the last row should be non-harvest to non-harvest.

6) Your statement that the data are available from the authors to academic researchers upon request does not satisfy this journal’s data policy as I understand it. You should make the data publicly available to all in an online repository.

Reviewer #2: 1. Originality and significance of contribution to existing knowledge

This paper investigates the impact of seasonal harvest timing on price convergence for staple grains across 97 Nigerian markets. The authors leverage weekly price data to argue that market integration is not merely a function of spatial factors but is critically mediated by the temporal synchrony of production cycles. Nevertheless, the empirical narrative largely confirms what the arbitrage condition already predicts (i.e., larger contemporaneous supplies compress gaps, asynchronous supplies widen them).

2. Appropriateness of the abstract as a description of the paper

The abstract clearly states the objective, dataset (weekly prices from 97 markets), and headline findings (faster convergence in synchronous harvests, slower in asynchronous ones). However, it provides no hint of the methodological framework.

3. Adequacy of the literature review

The authors successfully position their work within the existing literature on market integration, price transmission, and food security in Africa. The manuscript correctly references foundational and contemporary studies, effectively differentiating its temporal focus from prior work centered on spatial frictions. Yet the review leans heavily on sources published before 2020.

4. Organization and readability

The paper is logically structured, following a standard format that is easy to follow. The writing is generally clear. In Table 1, the variable "Average Price Gap (Harvest to Harvest)" is duplicated. Also, the authors reveals that the x-axis from Figure 1 is based only on the rainy season harvest, but does not explain why.

5. Soundness of methodology, analysis, and interpretation

The core methodological approach is suited to the research question. However, there are some concerns:

- the manuscript first states that “all data are fully available without restriction” and later that data “are available from the authors on request”; this is a contradiction;

- the authors acknowledge that their network data on market connections is not crop-specific, which introduces potential noise into the analysis;

- weekly prices offer granularity, yet the sample spans only 14 months (Feb 2022–Apr 2023); one harvest cycle could be insufficient to assess structural seasonal effects, macro shocks, or policy interventions;

- standard unit-root and cointegration tests should precede the regression;

6. Evidence supports conclusions.

The main conclusions are supported by the regression results. However, the paper lists no explicit limitation section.

6. PLOS authors have the option to publish the peer review history of their article (what does this mean?). If published, this will include your full peer review and any attached files.

Reviewer #1: No

Reviewer #2: No

---

## [Author Response · Author response to Decision Letter 1]

12 Sep 2025

Reviewer 1(R1):

This is a clearly-written and straightforward article using agricultural price data collected from 97 markets in Nigeria to show that locally-produced crops (rice and cowpea) experience faster price convergence across markets when both markets are in the harvest season. The analysis is based on reduced-form correlations. It would benefit from a better theoretical discussion of price behavior under storage costs (which explain low harvest prices that then increase throughout the year) and trade costs (e.g. under what conditions markets are segmented vs. trading) as this would help us understand the results and their implications. However, I think that is probably beyond the scope of this short article.

Response to Reviewer (RtoR):

We appreciate the reviewer’s suggestion to strengthen the theoretical discussion of price dynamics under storage and trade costs. In the revised manuscript, we have added a concise theoretical section that draws on the storage model (e.g., Williams and Wright, 1991; Wright, 2011) to explain how storage costs influence seasonal price patterns, and on standard trade cost frameworks (e.g., Samuelson, 1952) to clarify the conditions under which markets are segmented versus integrated. This additional discussion provides a clearer conceptual foundation for interpreting our empirical results, while keeping the focus and length appropriate for a short article.

R1:

Here are some minor comments that I think should be addressed:

1) Please provide more information on the markets. How were they selected? Can you show them on a map? How many are in the South and how many in the North? (This seems to be the key division in terms of your harvest seasons in table 5.)

RtoR:

We appreciate the reviewer’s suggestion. The revised manuscript now includes additional details on the selection of the 97 markets (Section 2: Page 3) and a new map (Figure 1) showing their locations. The map also indicates whether each market is in the North or South, as this distinction is important for understanding harvest seasonality. Of these markets, 61 are in the North and 36 are in the South.

R1:

2) You should reference the table 2 results in the main paper. One of the reasons why the Harvest to Non-Harvest price gaps appear larger in table 1 is that markets with different harvest seasons (North vs. South) are further apart.

RtoR:

Thank you for pointing this out. We have revised the manuscript to reference the Table 2 results in the main text and now explicitly note that part of the larger Harvest to Non-Harvest price gaps in Table 1 may reflect the greater distance between markets in the North and South. (last paragraph of Page 4)

R1:

3) P. 13 / table 3: if I am reading the table correctly, the “Harvest to Non-Harvest” category is NOT the slowest for White Cowpea as you state in the text. It is only the slowest for Rice.

RtoR:

Thank you for catching this inconsistency. You are correct—the “Harvest to Non-Harvest” category is the slowest only for rice. We have revised the text to accurately reflect the results for white cowpea in Table 3.

R1:

4) Appendix 1.a.: you should show the comparison that you did with the FEWS NET data. Also, please explain what is shown in figures 2 and 3 – is this your data or the FEWS NET data?

RtoR:

Thank you for the suggestion. We have updated Appendix 1.a to include the comparison with FEWS NET data. Figures 2 and 3 now clearly indicate their respective data sources—Figure 2 uses our market price survey data, while Figure 3 is based on FEWS NET prices. We have also added a brief discussion in the appendix explaining the observed patterns and noting that, despite differences in coverage and time span, the seasonal trends are broadly consistent, serving as a robustness check for our data.

R1:

5) Typos to fix: The first sentence of section II has a double “and”; in table 1 the last row should be non-harvest to non-harvest.

RtoR:

Thank you for pointing these out. We have corrected the double “and” in the first sentence of Section II and fixed the label in the last row of Table 1 to read “Non-Harvest to Non-Harvest.”

R1:

6) Your statement that the data are available from the authors to academic researchers upon request does not satisfy this journal’s data policy as I understand it. You should make the data publicly available to all in an online repository.

----

RtoR:

We appreciate this important clarification. In line with the journal’s data policy, we have now made both the dataset and replication code publicly available through an online repository on GitHub: [link]. We have updated the Data Availability Statement in the appendix to reflect this change.

Reviewer 2 (R2):

1. Originality and significance of contribution to existing knowledge

This paper investigates the impact of seasonal harvest timing on price convergence for staple grains across 97 Nigerian markets. The authors leverage weekly price data to argue that market integration is not merely a function of spatial factors but is critically mediated by the temporal synchrony of production cycles. Nevertheless, the empirical narrative largely confirms what the arbitrage condition already predicts (i.e., larger contemporaneous supplies compress gaps, asynchronous supplies widen them).

RtoR:

We appreciate the reviewer’s observation. Thank you for the summary.

R2:

2. Appropriateness of the abstract as a description of the paper

The abstract clearly states the objective, dataset (weekly prices from 97 markets), and headline findings (faster convergence in synchronous harvests, slower in asynchronous ones). However, it provides no hint of the methodological framework.

RtoR:

We thank the reviewer for pointing this out. We have revised the abstract to briefly mention the methodological framework, noting that our results are obtained from a dyadic panel regression of price-gap dynamics across market pairs classified by seasonal harvest alignment.

R2:

3. Adequacy of the literature review

The authors successfully position their work within the existing literature on market integration, price transmission, and food security in Africa. The manuscript correctly references foundational and contemporary studies, effectively differentiating its temporal focus from prior work centered on spatial frictions. Yet the review leans heavily on sources published before 2020.

RtoR:

We appreciate the reviewer’s positive assessment of our positioning within the literature. We agree that incorporating more recent studies would strengthen the review. Accordingly, we have added references to several relevant works published after 2020, particularly those addressing seasonal dynamics and market integration in African contexts.

R2:

4. Organization and readability

The paper is logically structured, following a standard format that is easy to follow. The writing is generally clear. In Table 1, the variable "Average Price Gap (Harvest to Harvest)" is duplicated. Also, the authors reveal that the x-axis from Figure 1 is based only on the rainy season harvest, but does not explain why.

RtoR:

We thank the reviewer for the positive comments on the paper’s structure and clarity. We have corrected the duplication of the “Average Price Gap (Harvest to Harvest)” variable in Table 1, this was a typo, which we have fixed. Regarding Figure 1(Figure 2 in our updated manuscript), we now clarify in the footnotes that the x-axis is anchored to the end of the rainy-season harvest to ensure consistency across markets, as the dry-season harvest occurs in fewer locations and roughly six months later.

R2:

5. Soundness of methodology, analysis, and interpretation

The core methodological approach is suited to the research question. However, there are some concerns:

- the manuscript first states that “all data are fully available without restriction” and later that data “are available from the authors on request”; this is a contradiction;

- the authors acknowledge that their network data on market connections is not crop-specific, which introduces potential noise into the analysis;

- weekly prices offer granularity, yet the sample spans only 14 months (Feb 2022–Apr 2023); one harvest cycle could be insufficient to assess structural seasonal effects, macro shocks, or policy interventions;

- standard unit-root and cointegration tests should precede the regression;

RtoR:

We thank the reviewer for these helpful observations. We have addressed each point as follows:

Data availability statement – We have resolved the inconsistency by making both the code and data publicly available via our GitHub repository [link] and updating the statement accordingly.

Network data crop-specificity – While the network links are not crop-specific, the crops we study are available in almost all markets (cowpea and local rice in all markets, long-grain rice in 90 of 97 markets). These links are only used to restrict the sample to connected markets and are not included as regressors. We acknowledge this as a potential source of noise and have explicitly noted it in the limitations section.

Sample period limitation – We acknowledge the short 14-month span and have added this to our limitations section, noting that future work using longer time series could better capture structural seasonal patterns, macroeconomic shocks, or policy interventions.

Unit-root and cointegration tests – We now report Maddala–Wu panel unit-root tests in the appendix and confirm that the price gap variables are stationary before running the regressions.

R2:

6. Evidence supports conclusions.

The main conclusions are supported by the regression results. However, the paper lists no explicit limitation section.

RtoR:

We thank the reviewer for this suggestion. We have now added an explicit Limitations section to the manuscript, outlining key constraints such as the non–crop-specific nature of the network data, the limited 14-month sample period, and the potential influence of unobserved local factors.

---

## [Decision Letter · Decision Letter 1]

19 Nov 2025

PONE-D-25-30518R1

Price Convergence in Grain Markets with Seasonal Differences

PLOS ONE

Dear Dr. Olabisi,

Thank you for submitting your manuscript to PLOS ONE. After careful consideration, we feel that it has merit but does not fully meet PLOS ONE’s publication criteria as it currently stands. Therefore, we invite you to submit a revised version of the manuscript that addresses the points raised during the review process. In particular pay attention to the comments of Reviewer #1 to equations, they are important to clearly identify your contribution.

We look forward to receiving your revised manuscript.

Kind regards,

Eugenio Sebastian Antonio Bobenrieth, Ph.D.

Academic Editor

PLOS ONE

Journal Requirements:

Reviewers' comments:

Reviewer's Responses to Questions

**Comments to the Author**

1. If the authors have adequately addressed your comments raised in a previous round of review and you feel that this manuscript is now acceptable for publication, you may indicate that here to bypass the “Comments to the Author” section, enter your conflict of interest statement in the “Confidential to Editor” section, and submit your "Accept" recommendation.

Reviewer #1: (No Response)

Reviewer #2: All comments have been addressed

2. Is the manuscript technically sound, and do the data support the conclusions?

Reviewer #1: Yes

Reviewer #2: Yes

3. Has the statistical analysis been performed appropriately and rigorously?

Reviewer #1: Yes

Reviewer #2: Yes

4. Have the authors made all data underlying the findings in their manuscript fully available?

Reviewer #1: Yes

Reviewer #2: Yes

5. Is the manuscript presented in an intelligible fashion and written in standard English?

Reviewer #1: Yes

Reviewer #2: Yes

6. Review Comments to the Author

Reviewer #1: Thank you for addressing my comments. I have some last minor points (numbers refer to my numbered comments on the previous version):

0) The new section on the Theoretical Framework is helpful. However, there are two points of confusion. First, line 62-63: This is not quite right. It is equation 1 that delivers the seasonal pattern by itself (prices are higher in non-harvest months because supply is coming from storage from the previous harvest). Equation 2 is just a reduced-form representation of the implications of equation 1. Equation 2 would make more sense if you present this more clearly as a 2-period model (harvest vs. non-harvest) rather than monthly. Prices are higher in the non-harvest period because of storage costs. Second, line 81-83: It was not clear what you meant by “domestic harvest enters weakly” as the domestic harvest does not appear in equation 5. Either include it explicitly in equation 5 or don’t mention it in the sentence. Also, why does equation 5 hold approximately and not with equality? It might also help to include another equation here after equation 5 that would be analogous to equation 4 but for the imported variety case.

1) The map is helpful but leaves me wondering two things. Why do some markets appear to have no links? And why are there no markets in Lagos and vicinity? It would be helpful to clarify in the text. Perhaps add more about the snowball approach or how the starting markets were selected.

2) This comment has been addressed.

3) This comment has been addressed.

4) This comment has been addressed.

5) This comment has been addressed.

6) This comment has been addressed.

Reviewer #2: In the first review report, some concerns were raised.

These concerns were successfully addressed by the authors in the revised manuscript.

The manuscript can be accepted in the present form.

7. PLOS authors have the option to publish the peer review history of their article (what does this mean?). If published, this will include your full peer review and any attached files.

Reviewer #1: No

Reviewer #2: **Yes: **Cristian Valeriu Stanciu

---

## [Author Response · Author response to Decision Letter 2]

25 Nov 2025

Reviewer 1:

Thank you for addressing my comments. I have some last minor points (numbers refer to my numbered comments on the previous version):

Reviewer Comment 0:

The new section on the Theoretical Framework is helpful. However, there are two points of confusion. First, line 62-63: This is not quite right. It is equation 1 that delivers the seasonal pattern by itself (prices are higher in non-harvest months because supply is coming from storage from the previous harvest). Equation 2 is just a reduced-form representation of the implications of equation 1. Equation 2 would make more sense if you present this more clearly as a 2-period model (harvest vs. non-harvest) rather than monthly. Prices are higher in the non-harvest period because of storage costs. Second, line 81-83: It was not clear what you meant by “domestic harvest enters weakly” as the domestic harvest does not appear in equation 5. Either include it explicitly in equation 5 or don’t mention it in the sentence. Also, why does equation 5 hold approximately and not with equality? It might also help to include another equation here after equation 5 that would be analogous to equation 4 but for the imported variety case.

Response to Reviewer Comment 0:

We are thankful for this hint from the reviewer. We have now revised the theoretical framework accordingly.

First, we now make it explicit that the storage no-arbitrage condition alone generates the seasonal pattern. To do so, we introduce a simple two-period example (harvest vs. non-harvest) and explain that prices are higher in the non-harvest period because supply comes from stored stocks and must cover storage costs.

Second, in the subsection on imported varieties, we removed the phrase “domestic harvest enters weakly” since harvest does not appear in that equation, and we now write the imported price relationship with equality rather than approximately. As you suggested, we also add a brief expression for the cross-market price gap for imports, directly analogous to the domestic-gap case. These changes address the points you raised. The updates can be found in line 64-76 and line 92-99.

Reviewer Comment 1:

The map is helpful but leaves me wondering two things. Why do some markets appear to have no links? And why are there no markets in Lagos and vicinity? It would be helpful to clarify in the text. Perhaps add more about the snowball approach or how the starting markets were selected.

Response to Reviewer Comment 1:

We are thankful for the guidance from the reviewer. We should start by admitting that Lagos was excluded largely because our limited budget could not cover it. Our snowball approach started from the largest known grain markets in the key production areas – predominantly in Northern Nigeria. These include Dawanau near Kano and Mutum Biu in Taraba State. Given our extensive coverage of Bodija market in Ibadan (about 70 miles from Lagos), and past research that shows the significance of Bodija to price movements in the southwest of Nigeria, we focused on the 97 markets used for this paper, with a reasonable level of confidence that our main findings, if we had the budget to include Lagos, would not be significantly different from our main findings as they are.

We are thankful that the reviewer used the phrase “appear to” in the question. We selected key link-markets that connect farms to consumers for our price surveys, but a few of these were not linked to other key markets – largely being linked to local food grain markets that are more than two steps removed from other key grain markets. Given that we could not track prices in all markets that we visited in 2022 - roughly 300 of them, there was no reason to plot a network graph that includes markets for which we have no price data. Furthermore, a network graph with 97 nodes is more legible and instructive than one with three times that number. In sum, the network graph reflects only linkages between key markets, specifically, the 97 markets for which we track prices.

We have now updated the paper to explain that we exclude Lagos because of budget limits, and why a few nodes in the graph appear to have no connections, because they had no reported links to other key grain markets that we track in our data.

The updates can be found near line 177-190 of the revised paper.

---

## [Editor Report · Decision Letter 2]

10 Dec 2025

Price Convergence in Grain Markets with Seasonal Differences

PONE-D-25-30518R2

Dear Dr. Olabisi,

We’re pleased to inform you that your manuscript has been judged scientifically suitable for publication and will be formally accepted for publication once it meets all outstanding technical requirements.

Kind regards,

Eugenio Bobenrieth

Academic Editor

PLOS One
---

## [Editor Report · Acceptance letter]

PONE-D-25-30518R2

PLOS One

Dear Dr. Olabisi,

I'm pleased to inform you that your manuscript has been deemed suitable for publication in PLOS One. Congratulations! Your manuscript is now being handed over to our production team.

Kind regards,

on behalf of

Dr. Eugenio Sebastian Antonio Bobenrieth

Academic Editor

PLOS One